# Mild Symptomatic SARS-CoV-2 P.1 (B.1.1.28) Infection in a Fully Vaccinated 83-Year-Old Man

**DOI:** 10.3390/pathogens10050614

**Published:** 2021-05-17

**Authors:** Marco Fabiani, Katia Margiotti, Antonella Viola, Alvaro Mesoraca, Claudio Giorlandino

**Affiliations:** 1Human Genetics Lab, ALTAMEDICA, Viale Liegi 45, 00198 Rome, Italy; marco.fabiani@artemisia.it (M.F.); antonella.viola@artemisia.it (A.V.); alvaro.mesoraca@artemisia.it (A.M.); claudio.giorlandino@artemisia.it (C.G.); 2Department of Biochemistry, ALTAMEDICA, Viale Liegi 45, 00198 Rome, Italy; 3Department of Prenatal Diagnosis, Fetal-Maternal Medical Centre, ALTAMEDICA, Viale Liegi 45, 00198 Rome, Italy

**Keywords:** COVID-19, immunity, SARS-CoV-2, P.1 variant, vaccine, Brazilian variant

## Abstract

The novel severe acute respiratory syndrome coronavirus (SARS-CoV-2) and the associated coronavirus disease 2019 (COVID-19) continue to spread throughout the world, causing more than 120 million infections. Several variants of concern (VOCs) have emerged and spread with implications for vaccine efficacy, therapeutic antibody treatments, and possible reinfections. On 17 March 2021, several VOCs were detected, including lineage B.1.1.7, first identified in the UK, B.1.351 in South Africa, Lineage P.1 (B.1.1.28.1) in Brazil, and novel Sub-Lineage A (A.23.1), reported in Uganda, and B.1.525, reported in Nigeria. Here, we describe an 83-year-old man infected with the SARS-CoV-2 P.1 variant after two doses of the BNT162b2 mRNA COVID-19 vaccine.

## 1. Introduction

The novel severe acute respiratory syndrome coronavirus (SARS-CoV-2) and the associated coronavirus disease 2019 (COVID-19) continue to spread throughout the world, causing more than 120 million infections. Several variants of concern (VOCs) have emerged and spread with implications for vaccine efficacy, therapeutic antibody treatments, and possible reinfections. Recently, a new SARS-CoV-2 lineage, P.1, was discovered in Manaus City, Amazonas, Brazil, in early January, 2020 [1,2]. The SARS-CoV-2 P.1 variant has a unique genetic profile of 17 amino acid changes, including three mutations (E484K, K417T, and N501Y) in the receptor-binding domain (RBD) region that are also present in the B.1.325 variant, a recent lineage that first emerged in South Africa [3]. It has been shown that B.1.351 is more resistant to neutralization by some monoclonal antibodies (mAbs), convalescent plasma, and vaccine sera, largely due to an E484K mutation that also exists in the P.1 lineage [1,4]. Of concern, in January 2021, the SARS-CoV-2 Lineage P.1 was able to cause a major second wave in Brazil [4]. In Italy, on 18 February, the prevalence of the Brazilian P.1 Lineage was 4.3%, ranging between 0% and 36.2% in different regions (ISS, second survey on the prevalence of variants of the Sars-CoV-2 virus (salute.gov.it, accessed on 19 March 2021)). This report describes an infective episode with SARS-CoV-2 Lineage P.1 occurring in an individual in Italy after the second dose of the Pfizer BNT162b2 mRNA COVID-19 vaccine [5].

## 2. Case Presentation

Here, we describe the case of an 83-year-old male who, on 16 January 2021 received the first dose of the Pfizer BNT162b2 mRNA COVID-19 vaccine, receiving the second dose on 16 February and completing the vaccination program. He received the vaccine in Santiago de Chile, South America, and he traveled back to Italy on 21 February. On 11 March, 2021 he was in close contact with a work colleague who had COVID-19—both the 83-year-old male and his work colleague are employed and live in Rome, Italy. On 17 March, we processed the 83-year-old male’s nasopharyngeal swab at the Altamedica Laboratory for SARS-CoV-2 detection by real-time RT-PCR, obtaining a positive result. His PCR test was found to be positive with cycle threshold (CT) values of the ORF1ab region, E and N genes 12, 13, and 12. To characterize the SARS-CoV-2 virus lineage, we performed a screening step of genotyping RT-PCR to detect the main SARS-CoV-2 mutations of the VOCs. This step was performed with custom TaqMan probes directed toward the following mutations: D1118H, T716I, S982A, del69-70, P681H, D614G, A570D, N501Y, del144, K417T, E484K, D215G, A701V, D138Y, T1027I. Once we detected the specific mutations of the P.1 variant (N501Y, K,417T, E484K, D138Y, D614G, and T1027I), we proceeded to perform next-generation sequencing (NGS) to confirm the lineage and investigate further mutations. In Figure 1 and Table 1, we report the unique amino acid changes revealed in the patient by NGS. In addition to the reported mutation in the GSAID database for the P.1. variant (S1188L, K1795Q, S3675-F3677del for orf1ab gene, L18F, T20N, P26S, D138Y, R190S, K417T, E484K N501Y, H655Y, T1027I, D614G for S gene, G174C for orf3a, E92K for orf8, and P80R for N gene), further missense mutations and variants with neutral effects have also been detected (Table 1). Among the missense mutations, we detected A1809V, V4225L, P4715L, D5130Y, and E5665D in the orf1ab gene, S640F and V1176F in the S gene, and RG203KR in the N gene. The result of Pangolin tool submission was, as expected from previous RT-PCR results, the Brazilian P.1 Lineage, a new and recently identified SARS-CoV-2 variant [3].

On 19 March, laboratory analysis showed a positive serologic test for the antibody anti-spike protein IgG, and total antibody anti-S1 SARS-CoV-2 virus, while the patient had a negative test for IgM and IgA antibodies (Table 2). Three days after a positive COVID-19 diagnosis, the patient had mild symptoms, such as headache and a cold. The symptoms were managed at home and resolved in two days. On 22 March, the patient repeated the nasopharyngeal swab, obtaining a further positive SARS-CoV-2 result with a significantly reduced viral load (ORF1ab CT 30, E gene CT 32, and N gene CT 31). Four days later, on 26 March, the PCR test was completely negative for all three genes (Figure 2).

## 3. Discussion

The spread of new SARS-CoV-2 variants represents a serious threat worldwide and, recently, it has been shown that an artificially created virus (pseudovirus) carrying the same mutations as the Brazilian P.1 variant also escaped from anti-receptor-binding domain (RBD) monoclonal antibodies and sera or plasma from convalescent or Pfizer BNT162b2-vaccinated individuals [6]. Another study showed that the immune plasma of COVID-19 convalescent blood donors had six-fold less neutralizing capacity against the P.1 lineage with respect to the B lineage [7]. Here, we described a fully BNT162b2-vaccinated 83-year-old man infected with the Brazilian P.1 SARS-CoV-2 variant. People are considered fully vaccinated for COVID-19 two weeks after they have received the second dose in a two-dose series (Pfizer–BioNTech). He had no symptoms, such as fever, cough, or shortness of breath, only experiencing a slight headache and mild cold. On 26 March, the PCR test was completely negative for all three genes, 10 days after his first positive PCR test, showing a shorter positive period with respect to the mean of infected patients. NGS analysis, besides confirming all the genetic variations typical of the P.1 lineage, identified other missense and neutral mutations, as shown in Table 1. Among them, we found the P4715L mutation mapped in the ORF1ab region, which a recent study found to be associated with a higher mortality rate for COVID-19 when linked to the S protein D614G mutation [8], also identified in our patient (Table 1). Similarly, we found missense mutation V1176F, present in the spike protein region, which is also correlated with higher mortality ratios and increased fitness of SARS-CoV-2 infection [9]. A qualitative serology test revealed the presence of IgG, and a quantitative serology test revealed the presence of 319.0 BAU/mL of total anti-spike SARS-CoV2 virus antibody. The serology test was carried out five weeks after the second dose of the Pfizer–BioNTech vaccine after the positive SARS-CoV-2 virus PCR test but two days before mild symptom onset. According to the literature reports, antibodies are usually detectable approximately 4-6 days after PCR confirmation of infection, suggesting that the revealed antibody response was mostly vaccine induced, rather than a natural immunity response to the virus. It has been estimated that the Pfizer–BioNTech vaccine has 90% effectiveness after the second dose in individuals aged 80 years or older, and at least 97% effectiveness against symptomatic COVID-19 cases, hospitalizations, and deaths [10]. Poor neutralization activity against the P.1 variant has been reported, mainly related to the E484K mutation alone or in combination with other mutations in the receptor-binding domain, reducing the vaccine efficiency, so fully vaccinated people should be mindful of the potential risk of becoming infected and transmitting the virus to other people [11,12,13].

What it is important to underline here is that, in this case, despite being 83 years old, the patient had almost no symptoms and completely recovered from COVID-19 disease after a few days, unlike his younger colleagues. In his place of work, a small outbreak occurred, infecting 19 out of 21 colleagues. In particular, four of them, with an age ranging from 54 to 61 years old, showed severe COVID-19 symptoms, resulting in bilateral pneumonia and oxygen supplementation and, for one of them, in hospitalization and intensive care unit (ICU) admission. Interestingly, no coworkers were vaccinated, and they had never had a positive COVID-19 test before. Notably, the seventy-five-year-old wife of the infected patient was vaccinated and at the time of her husband’s infection, she had an anti-spike SARS-CoV2 virus antibody titer of 1305 BAU/mL, four-fold higher than her husband, and although she lives in direct contact with him, she never contracted SARS-CoV-2 infection. As previously described in studies about SARS-CoV-2-positive patients and those who have recovered, neutralizing antibodies can provide protection from infection and attenuate disease symptoms [14,15]. Since the investigated patient is a frequent traveler often underwent to SARS-CoV-2 swabs diagnostic test, and he did not refer cold like symptoms since pandemic outbreak, so we can exclude that he was previously infected by the virus. In conclusion, the 83-year-old man described in this study as being infected with the Brazilian P.1 SARS-CoV-2 variant had mild symptoms, which resolved in only two days (Figure 2), so we can speculate that, regardless of the reported reduced efficacy of the Pfizer–BioNTech vaccine against the P.1 Brazilian variant, the vaccine-induced immune response attenuates the disease symptoms, which are often moderate to severe for >80-year-old individuals.

## 4. Materials and Methods

### 4.1. Detection of the SARS-CoV-2 Virus by Real-Time Polymerase Chain Reaction (RT-PCR)

Specimens were obtained from the patient by nasopharyngeal swab and blood collection. Nucleic acid extraction was performed using the automated PANA 9600s extractor system. Detection of the SARS-CoV-2 RNA was performed by RT-PCR (KHB, diagnostic kit for SARS-CoV-2). The assay targets three genes of SARS-CoV-2 (ORF1ab, N, and E).

### 4.2. SARS-CoV-2 Variant Characterization by RT-PCR and Next-Generation Sequencing (NGS)

An aliquot of extracted RNA underwent a retrotranscription (RT) step. cDNA was synthesized from the extracted RNA using the SuperScript IV VILO Master Mix (Thermo Fisher Scientific, Waltham, MA, USA) according to the manufacturer’s instructions. The first step of RT-PCR was performed for the detection of the main SARS-CoV-2 mutations of the three variants. This step was performed with custom TaqMan probes directed at the following mutations: D1118H, T716I, S982A, del69-70, P681H, D614G, A570D, N501Y, del144, K417N, E484K, D215G, A701V, D138Y, T1027I. The sequencing analysis of SARS-CoV-2 was performed using an Ampliseq SARS-CoV-2 panel (TermoFisher Scientific). The reads from the library were aligned with the Wuhan-Hu-1 NCBI Reference Genome (Accession Number: MN908947.3) in Torrent Suite v. 5.10.1. For mutation calling, the following plugins were used: Coverage Analysis (v. 5.10.0.3), Variant Caller (v.5.10.1.19), and COVID19AnnotateSnpEff (v.1.0.0), a plugin specifically developed for SARS-CoV-2 that can predict the effect of a base substitution [16]. The software Integrative Genomics Viewer_2.8.0 (IGV) was used to visualize the torrent variant caller (TVC) bam files of each sample in order to check the consistency of nucleotide calls. Raw sequence reads were aligned to the complete genome of the SARS-CoV-2 Wuhan-Hu-1 isolate (Genbank accession number: NC_045512.2) and classified using the Pangolin COVID-19 Lineage Assigner tool v. 2.0.7 (http://pangolin.cog-uk.io/, accessed on 19 March 2021) [17].

## Figures and Tables

**Figure 1 pathogens-10-00614-f001:**
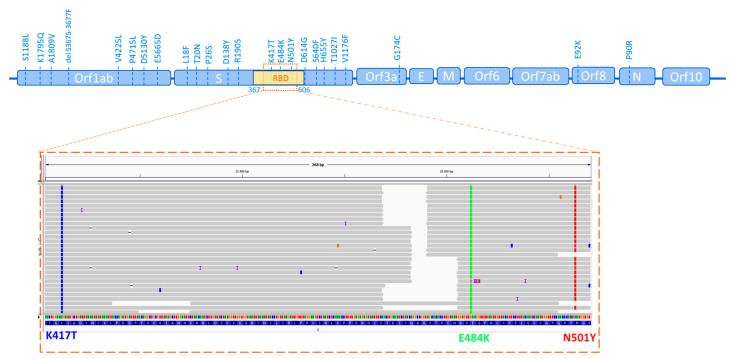
Schematic representation of mutation of analyzed patients’ SARS-CoV-2 with an aminoacidic missense change revealed by NGS. Genome IGV analysis that shows the three characterizing P.1 mutations in the reads from sequenced samples.

**Figure 2 pathogens-10-00614-f002:**
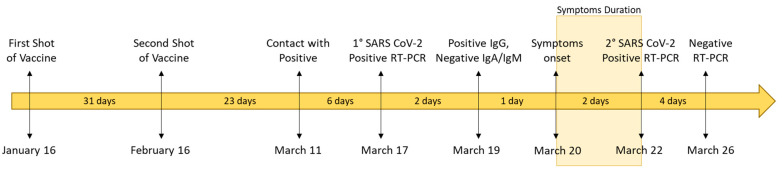
Timeline of vaccine, symptom onset, and molecular diagnosis of patient.

**Table 1 pathogens-10-00614-t001:** Number of gene mutations in SARS-CoV-2 genome of patient.

Position	Gene	HGVS_C	HGVS_P	Mutation Effect
733	orf1ab	c.468T > C	D156D	synonymous mutation
2749	orf1ab	c.2484C > T	D828D	synonymous mutation
3037	orf1ab	c.2772C > T	F924F	synonymous mutation
3828	orf1ab	c.3563C > T	S1188L	missense mutation
5648	orf1ab	c.5383A > C	K1795Q	missense mutation
5691	orf1ab	c.5426C > T	A1809V	missense mutation
6319	orf1ab	c.6054A > G	P2018P	synonymous mutation
6613	orf1ab	c.6348A > G	V2116V	synonymous mutation
11287	orf1ab	c.11023_11031del	S3675_F3677del	conservative inframe deletion
12778	orf1ab	c.12513C > T	Y4171Y	synonymous mutation
12938	orf1ab	c.12673G > T	V4225L	missense mutation
13851	orf1ab	c.13587T > C	G4529G	synonymous mutation
13860	orf1ab	c.13596C > T	D4532D	synonymous mutation
14408	orf1ab	c.14144C > T	P4715L	missense mutation
15652	orf1ab	c.15388G > T	D5130Y	missense mutation
17259	orf1ab	c.16995G > T	E5665D	missense mutation
26149	orf3a	c.757T > C	S253P	missense mutation
21614	S	c.52C > T	L18F	missense mutation
21621	S	c.59C > A	T20N	missense mutation
21638	S	c.76C > T	P26S	missense mutation
21974	S	c.412G > T	D138Y	missense mutation
22006	S	c.444C > T	N148N	synonymous mutation
22132	S	c.570G > T	R190S	missense mutation
22812	S	c.1250A > C	K417T	missense mutation
23012	S	c.1450G > A	E484K	missense mutation
23063	S	c.1501A > T	N501Y	missense mutation
23403	S	c.1841A > G	D614G	missense mutation
23481	S	c.1919C > T	S640F	missense mutation
23525	S	c.1963C > T	H655Y	missense mutation
24642	S	c.3080C > T	T1027I	missense mutation
25088	S	c.3526G > T	V1176F	missense mutation
28167	orf8	c.274G > A	E92K	missense mutation
28512	N	c.239C > G	P80R	missense mutation
28881	N	c.608_610delGGGinsAAC	RG203KR	missense mutation

**Table 2 pathogens-10-00614-t002:** Patient characteristics and laboratory test results.

Patient	Characteristics
Age	83
Sex	Male
Symptoms	Headache and cold
Laboratory results	
Sars-CoV-2 IgG	Positive
Sars-CoV-2 IgM	Negative
Sars-CoV-2 IgA	Negative
Sars-CoV-2 Ab anti-spike (RBD)	319.0 BAU/mL

## Data Availability

The data presented in this study are available on request from the corresponding author.

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
