# Peer review of "Mild Symptomatic SARS-CoV-2 P.1 (B.1.1.28) Infection in a Fully Vaccinated 83-Year-Old Man"

_pathogens, 2021, doi:10.3390/pathogens10050614_

Round 1

Reviewer 1 Report

In the manuscript "Mild symptomatic SARS-CoV-2 P1 (B.1.1.28) infection in a fully vaccinated 83 years old man" the Authors present an interesting case report of mild COVID-19 in the course of infection with a Brazilian variant of SARS-CoV-2 in the patient who received two doses of mRNA vaccine. The paper is well written, however, there is limited novelty and predictable outcomes. It is well known that available vaccines do not fully protect against the infection especially in case of a variant of concerns. There are also some minor comments. Why the patient was diagnosed for SARS-CoV-2 infection before the clinical symptoms occurred? I understand that it was caused by contact with an infected work colleague. Since you do not present the genomic sequencing of SARS-CoV-2 in the infected colleague that cannot be proved that infection was acquired this way. Please consider the submitting of the manuscript as a "Letter to Editor".

Author Response

Reviewer :

In the manuscript "Mild symptomatic SARS-CoV-2 P1 (B.1.1.28) infection in a fully vaccinated 83 years old man" the Authors present an interesting case report of mild COVID-19 in the course of infection with a Brazilian variant of SARS-CoV-2 in the patient who received two doses of mRNA vaccine. The paper is well written, however, there is limited novelty and predictable outcomes. It is well known that available vaccines do not fully protect against the infection especially in case of a variant of concerns. There are also some minor comments. Why was the patient diagnosed for SARS-CoV-2 infection before the clinical symptoms occurred? I understand that it was caused by contact with an infected work colleague. Since you do not present the genomic sequencing of SARS-CoV-2 in the infected colleague that cannot be proved that infection was acquired this way. Please consider the submitting of the manuscript as a "Letter to Editor".

Answer

Thank you for your revision, it is known that vaccines do not fully protect against the infection, but it is still not so clear in the real world the efficiency of the vaccine in terms of protection against the severe form of COVID-19 disease especially referred to this new VOCs. We wanted to show a particular clinical case of SARS-CoV-2 P.1 (B.1.1.28) infection in a fully vaccinated 83-year-old patient in which the SARS-CoV-2 virus infection was harmless. Since there are a few cases regarding infection of vaccinated people with this new VOCs we considered of interest to share the information with the scientific community. The patient was recommended to do nasopharyngeal swab for SARS-CoV-2 detection since a coworker was positive. We do not have SARS-CoV-2 sequence of the first contact because this one performed the analysis in a different laboratory. But on the other hand, we performed the analysis on all the other colleagues that came in contact with him, and all resulted positive for the SARS-CoV-2 Brazilian variant, so likely the first contact would have the same variant as well. Regarding your kind suggestion to submit our manuscript as a "Letter to Editor", since the magnitude of our data and the few articles available regarding the P.1 variant response to vaccine, we considered  that “Case Report” type of article was more suitable.

Reviewer 2 Report

The presented MS describes a case report on a single male, fully vaccinated octogenarian who nevertheless was infected by SARS-CoV, approximately a month after the second shot and developed for two mild symptom COVID-19.

It is well elaborated, but poorly presented regarding the quality of its English. I have no major critics of the MS apart of the English. Word use, orthography and grammar are absolutely deficient. There is such an enormous amount of flaws and small errors that it turns the MS in parts almost unreadable. A minimum linguistic standard has to be met if the authors want to publish their work as scientific publication in an international journal. 
Just two example: Articles, definite as well as indefinite ones, exist in the English language and they are there to use them, in the correct way, of course. Spelling mistakes: "revels" has another meaning than "reveals", "manly" is something different than "mainly".
Altogether I have the impression that the MS at least in parts was hastily written. Somewhat more care results usually helpful for a scientific publication.

I recommend major revision of the English of the WHOLE text. Revise the whole text very carefully with the help of a native speaker.
Below, a list of small errors including some linguistic ones. The latter are by far not all.

Some comments regarding the Discussion section:
Consider structuring.
l 82 Define/explain "pseudovirus".

l 85-86 Six-fold less compared to what o whom, the wt = Wuhan lineage? Explain clearly.

The stunning fact about this case is that despite vaccination and therefore active immunization a month later the individual developed a SARS-CoV-2 infection of moderate degree. As far as I understand an antibody titer of 319 BAU/ml can not be considered as high. It might be suggested that this low degree of serum antibody concentration, aprox. 1 month after the second vaccine shot, meant a state of only partial protection. However, absence of IgM/presence of IgG abs indicate that the immune response was no longer in its initial phase. Authors could discuss if this was the reason that the immune protection failed and the infection could develop to mild symptoms.

Authors should mention (and discuss) if the individual was previously (in 2020, before vaccination) infected with SARS-CoV-2.

Formal errors
l 9: Should read: "*  These authors contributed equally".
l 10: Should read: "Y  Correspondence: Katia Margiotti, ..."

Left-side citation hint: I wonder that the last author, C. Giorlandino, suddenly appears first in "Citation: Giorlandino, C.; Lastname, F; Lastname, F. Title Pathogens 2021, 10, x. https://doi.org/10.3390/xxxxx". Correct as necessary.

References:
Only for one reference a DOI is given although many medRxiv articles and other electronic sources are cited that use to have DOIs. DOIs for the majority of references have to be provided, but at least for all internet-accesible works.

Fig 1 Indicate in the upper graph the location of the RBD or alternatively mention its size in the text, aprox. start to end position.

Fig 2 The figure is not on scale, I guess intentionally. For the reader it would be helpful to have indicated in the "Timeline" the number of days, counted either from the beginning or from the day of the Second vaccine shot (as - and + days).

Linguistic shortcomings:
A couple of mistakes are listed below, but by far these are not all. I stopped to take notes when realizing that they are too numerous.
l. 35:  Missing "It" at beginning of phrase.
l. 37 Missing "the" before "P.1 lineage".
l 39. the range 0 to 36.2% is given twice. Either explain or remove once.
l 42: "injection" missing after "second dose"

l 48 Replace "in" by "to".
l 49-50: Phrase is not comprehensible. Rephrase and write in understandable English, PLEASE.
l 53: What should mean "mains"? Again: English, please.

Multiple times throuout the Discussion section: "variant" was used falsely instead of "mutation".
l 95/97: "P4715L/D614G variant" should read "P4715L/D614G mutation", respectively. Check in whole paragraph.

l 137: direct or directed? Btw, probes instead of probe.
l 145: Integrative Genomics Viewer

In my opinion it makes no sense to do only good science when the efforts are later not presented to the global scientific community in a way that the results and their meaning is comprehensible to everyone. I have no doubt that the authors share this opinion.

Author Response

Reviewer 

“The presented MS describes a case report on a single male, fully vaccinated octogenarian who nevertheless was infected by SARS-CoV, approximately a month after the second shot and developed for two mild symptom COVID-19.It is well elaborated, but poorly presented regarding the quality of its English. I have no major critics of the MS apart of the English. Word use, orthography and grammar are absolutely deficient. There is such an enormous amount of flaws and small errors that it turns the MS in parts almost unreadable. A minimum linguistic standard has to be met if the authors want to publish their work as scientific publication in an international journal. 
Just two example: Articles, definite as well as indefinite ones, exist in the English language and they are there to use them, in the correct way, of course. Spelling mistakes: "revels" has another meaning than "reveals", "manly" is something different than "mainly".
Altogether I have the impression that the MS at least in parts was hastily written. Somewhat more care results usually helpful for a scientific publication. I recommend major revision of the English of the WHOLE text. Revise the whole text very carefully with the help of a native speaker.
Below, a list of small errors including some linguistic ones. The latter are by far not all.”

“ Some comments regarding the Discussion section:
Consider structuring. “l 82 Define/explain "pseudovirus".”

Answer :Pseudovirus definition was added at Line 91 in Discussion section.

“l 85-86 Six-fold less compared to what o whom, the wt = Wuhan lineage? Explain clearly.”

Answer :New sentence was added at Lines 95-96 in Discussion section

“The stunning fact about this case is that despite vaccination and therefore active immunization a month later the individual developed a SARS-CoV-2 infection of moderate degree. As far as I understand an antibody titer of 319 BAU/ml cannot be considered as high. It might be suggested that this low degree of serum antibody concentration, aprox. 1 month after the second vaccine shot, meant a state of only partial protection. However, absence of IgM/presence of IgG abs indicate that the immune response was no longer in its initial phase. Authors could discuss if this was the reason that the immune protection failed, and the infection could develop to mild symptoms.“

Answer : As today, it is not possible to define an antibody titer threshold of protection. Given the specificity of the used kit we can conclude only that an antibody titer > 50 BAU/ml can be considered as positive, but due to the lack of retrospective or observational studies in the field as well as the lack of standardization in measuring antibody titer we cannot certainly predict the effectiveness of vaccine protection using only antibody titer. Anyway, we added in the manuscript addition information regarding the antibody titer of the patient’s wife to better speculate on the possibility that her higher antibodies titer cloud be more protective against infection. (Lines 131-135 Discussion section).

“Authors should mention (and discuss) if the individual was previously (in 2020, before vaccination) infected with SARS-CoV-2.”

Answer : Thank you for the suggestion. We include patient’s information in the Manuscript. Line 137-140 in Discussion section.

“Formal errors.

l 9: Should read: "*  These authors contributed equally".”

Answer: Corrected

“l 10: Should read: "Y  Correspondence: Katia Margiotti, ..."”

Answer: Corrected

“ Left-side citation hint: I wonder that the last author, C. Giorlandino, suddenly appears first in "Citation: Giorlandino, C.; Lastname, F; Lastname, F. Title Pathogens 2021, 10, x. https://doi.org/10.3390/xxxxx". Correct as necessary.”

Answer: Corrected

“References:
Only for one reference a DOI is given although many medRxiv articles and other electronic sources are cited that use to have DOIs. DOIs for the majority of references have to be provided, but at least for all internet-accesible works”.

Answer: Thank you to made us notice this issue. There was a problem with reference software updating; DOI link as well as chapters and pages of reference was added.

Fig 1 Indicate in the upper graph the location of the RBD or alternatively mention its size in the text, aprox. start to end position.

Answer: Thanks for your suggestion we edited the Fig 1 following your instruction

Fig 2 The figure is not on scale, I guess intentionally. For the reader it would be helpful to have indicated in the "Timeline" the number of days, counted either from the beginning or from the day of the Second vaccine shot (as - and + days).

Answer: Thanks for your suggestion we improved the Fig 2 following your instruction

“Linguistic shortcomings:
A couple of mistakes are listed below, but by far these are not all. I stopped to take notes when realizing that they are too numerous.
l. 35:  Missing "It" at beginning of phrase.
l. 37 Missing "the" before "P.1 lineage".
l 39. the range 0 to 36.2% is given twice. Either explain or remove once.
l 42: "injection" missing after "second dose"
l 48 Replace "in" by "to".
l 49-50: Phrase is not comprehensible. Rephrase and write in understandable English, PLEASE.
l 53: What should mean "mains"? Again: English, please.”

Answer: Corrected

“Multiple times throughout the Discussion section: "variant" was used falsely instead of "mutation".
l 95/97: "P4715L/D614G variant" should read "P4715L/D614G mutation", respectively. Check in whole paragraph.”

Answer: Thank you for your suggestion, the word “variant” was replaced in “mutation” in all the MS. In literature variants and mutations are likewise used, anyway, in this MS , as you suggested us, it is more appropriate to use the word mutations instead of variant to avoid misunderstanding with SARS-Cov2 variants.

“l 137: direct or directed? Btw, probes instead of probe.
l 145: Integrative Genomics Viewer.”

Answer: Corrected

“In my opinion it makes no sense to do only good science when the efforts are later not presented to the global scientific community in a way that the results and their meaning is comprehensible to everyone. I have no doubt that the authors share this opinion.”

Answer: Thank you for all your English grammar correction and your expert scientific advice. Since the large amount of grammar errors, as you can notice we requested an extensive English editing service. We believe that thanks to you and the editing service the manuscript has become much more readable and understandable.

Reviewer 3 Report

The manuscript is interesting and well-prepared; indeed, I learned from reading it.  If the information is available, it would be interesting to know the vaccine and / or prior infection status of the coworkers who were infected -- at least those who had more severe disease that required admission to the hospital or ICU.

Author Response

Reviewer

“The manuscript is interesting and well-prepared; indeed, I learned from reading it.  If the information is available, it would be interesting to know the vaccine and / or prior infection status of the coworkers who were infected -- at least those who had more severe disease that required admission to the hospital or ICU.”

Answer : Thank you for the suggestion. We include requested information in the Manuscript. Line 127-132 in Discussion section

Round 2

Reviewer 1 Report

The Authors explanation is satisfactory.

Reviewer 2 Report

The authors responded to all my comments in a appropriate manner. The MS improved and, as authors state, is much more readable and understandable. The English is now up to common standards for scientific publication.